# Depression, GABA, and Age Correlate with Plasma Levels of Inflammatory Markers

**DOI:** 10.3390/ijms20246172

**Published:** 2019-12-06

**Authors:** Amol K. Bhandage, Janet L. Cunningham, Zhe Jin, Qiujin Shen, Santiago Bongiovanni, Sergiy V. Korol, Mikaela Syk, Masood Kamali-Moghaddam, Lisa Ekselius, Bryndis Birnir

**Affiliations:** 1Department of Neuroscience, Physiology, Uppsala University, BMC, Box 593, 75124 Uppsala, Sweden; Amol.Bhandage@su.se (A.K.B.); Zhe.Jin@neuro.uu.se (Z.J.); Sergiy.Korol@neuro.uu.se (S.V.K.); 2Department of Neuroscience, Psychiatry, Uppsala University, 75185 Uppsala, Sweden; Janet.Cunningham@neuro.uu.se (J.L.C.); Santiago.Bongiovanni@akademiska.se (S.B.); Mikaela.Syk@neuro.uu.se (M.S.); Lisa.Ekselius@neuro.uu.se (L.E.); 3Department of Immunology, Genetics and Pathology, Science for Life laboratory, Uppsala University, 75108 Uppsala, Sweden; Qiujin.Shen@igp.uu.se (Q.S.); Masood.Kamali@igp.uu.se (M.K.-M.)

**Keywords:** GABA_A_ receptor, inflammation, mental health

## Abstract

Immunomodulation is increasingly being recognised as a part of mental diseases. Here, we examined whether levels of immunological protein markers changed with depression, age, or the inhibitory neurotransmitter gamma-aminobutyric acid (GABA). An analysis of plasma samples from patients with a major depressive episode and control blood donors (CBD) revealed the expression of 67 inflammatory markers. Thirteen of these markers displayed augmented levels in patients compared to CBD. Twenty-one markers correlated with the age of the patients, whereas 10 markers correlated with the age of CBD. Interestingly, CST5 and CDCP1 showed the strongest correlation with age in the patients and CBD, respectively. IL-18 was the only marker that correlated with the MADRS-S scores of the patients. Neuronal growth factors (NGFs) were significantly enhanced in plasma from the patients, as was the average plasma GABA concentration. GABA modulated the release of seven cytokines in anti-CD3-stimulated peripheral blood mononuclear cells (PBMCs) from the patients. The study reveals significant changes in the plasma composition of small molecules during depression and identifies potential peripheral biomarkers of the disease.

## 1. Introduction

Neurotransmitter signaling in the nervous system has been well-studied, where gamma-aminobutyric acid (GABA) is the main inhibitory transmitter [1]. Compelling evidence demonstrates that neurotransmitter signaling also takes place in the immune system [2,3,4,5,6,7]. The fact that cross-talk occurs between the immune and nervous systems is not surprising. It may be required for normal brain functions and is probably essential for coordinated stress, emotional, and behavioral responses [8]. Dysregulation of the immune system has furthermore been reported to be associated with psychiatric disorders, such as depression [8]. Pro-inflammatory cytokines can induce sickness behavior that resembles major depressive disorder (MDD) and interferon-alpha (INF-α) treatment induces MDD in about 25% of cases, suggesting causal mechanisms [9,10]. Pro-inflammatory markers such as IL-6, IL-1β, IFN-α, TNF-α, and MCP-1/CCL2 are increased in the blood and cerebrospinal fluid (CSF) from patients with mood disorders compared to healthy controls when assessed at the baseline and also after exposure to stressors [11,12,13]. Inflammatory markers such as IL-6 and C-reactive protein (CRP) are consistently found to be elevated in depression, although the size of the effect is relatively small [14,15]. Emerging evidence also indicates that antidepressants have immunomodulating effects and that inflammatory and pro-inflammatory cytokines undermine the treatment response to conventional antidepressants [16,17,18,19,20,21]. Understanding the immunological changes in depression is important, as immunomodulation may be a possible therapy for some patients with depression.

In the brain, GABA is produced from glutamate in neuronal cells by the enzyme glutamic acid decarboxylase (GAD) [22]. Central nervous system (CNS) interstitial GABA and the human plasma GABA concentrations are expected to be in the submicromolar range [23,24,25,26], though the origin of GABA in blood is still being explored. A recently identified drainage system of the brain, the glymphatic system, indicates that the brain is a significant source of the GABA present in blood [27]. The expression of GABA receptor subunits and activation of functional GABA_A_ receptors have been recorded in immune cells such as peripheral blood mononuclear cells (PBMCs), T cells, monocytes, dendritic cells, and macrophages [7,28]. Recently, we demonstrated that GABA inhibits the secretion of a variety of inflammatory protein markers from PBMCs and T cells from healthy individuals and type 1 diabetes patients [7]. Nevertheless, the effects of GABA on the secretion of cytokines/immunological markers from immune cells is still relatively unexplored.

Here, we analyzed the immunological markers in plasma from control blood donors (CBD) and patients (PD) with a major depressive episode, and examined whether the levels of markers changed with age. We further studied the expression of GABA signaling system components and the effects of GABA treatment on the inflammatory marker profile of stimulated PBMCs from the patients. The results highlight augmented levels of immunological markers and the neurotransmitter GABA in the plasma of patients, together with altered GABA signaling in PBMCs from patients. The results are consistent with the immunomodulatory effects of GABA during depression. Interestingly, the level of a number of inflammatory markers correlated with age for both groups.

## 2. Results

### 2.1. Demographic Data

Demographic data for the individuals (CBD:26; PD:25) that participated in the study are shown in Table 1 and Appendix A. In total, 38 patients that met the criteria were selected for this study. Of these 38 patients, seven patients chose not to participate, while six individuals were unable to provide informed consent due to cognitive symptoms. Therefore, 25 patients were included in the study (Table 1). All the patients met the Diagnostic and Statistical Manual of Mental Disorders (DSM)-IV criteria for a current, moderate to severe depressive episode and either major depressive disorder or bipolar disorder. They were all undergoing treatment for depression at the Department of General Psychiatry at Uppsala University Hospital, Sweden, at the time the samples were obtained. Eleven of the patients were prescribed benzodiazepines or “Z-drugs”, while three of the patients had both. None of the patients had a documented history of alcohol addiction or abuse disorder and none had consumed alcohol during the week prior to the sampling. Two patients had received electroconvulsive therapy (ECT) during the past three months, but none had received it during the past month. Five patients had previously received ECT during their lifetime. Two patients had neurodevelopmental disorders, but the physician evaluated them to be capable of judgment in terms of giving consent. In three cases, the MINI interview could not be performed due to cognitive symptoms; in one case, the patient developed psychotic symptoms with delusions and severe disorganized thinking 24 h after giving consent, another patient presented severe concentrating difficulties, and the last one did not consent to the interview due to fatigue. Diagnosis in these cases was made based on clinical records. CBD gave blood at the blood center at Uppsala University Hospital and were age- and gender-matched to the patients, but were not evaluated in terms of mental health.

### 2.2. Inflammatory Markers in Plasma from Patients and CBD

Immune cells release a large number of small proteins, collectively called inflammatory markers, which may have a protective function or act as pro-inflammatory molecules. We investigated whether the types of inflammatory markers in plasma differed between CBD and the patients. We measured the plasma levels of 92 inflammatory markers that are most commonly associated with inflammation using an Olink inflammation panel analyzed with a multiplex proximity extension assay (PEA) (Appendix A) (http://www.olink.com/products/inflammation/#). The technology uses paired antibodies for the different inflammatory markers, such as cytokines, growth factors, mitogens, chemotactic, soluble receptors, and other pro-inflammatory molecules, and this allows a comparison of the levels of the same marker in samples from, e.g., CBD and the patients. However, the assay format does not support a comparison of the absolute levels of one marker to another as the affinities of the antibodies for their cognate targets may vary. In plasma from both CBD and the patient group, 67 inflammatory markers out of 92 analyzed proteins were detectable with values above the limit of detection (LOD) (Figure 1A; Appendix A). Importantly, 13 inflammatory markers were significantly higher in plasma from the patients compared to markers in plasma from CBD (Figure 1B).

### 2.3. Effects of Age on Levels of Inflammatory Markers in Plasma

Since the patients varied in age, we examined if there were any correlations between age and the level of inflammatory markers in plasma from the two groups. Ten inflammatory markers correlated with age in CBD (Figure 2A; Appendix A) and 21 in the patients (Figure 2B; Appendix A). Six inflammatory markers, including IL-8, CXCL9, HGF, VEGF-A, OPG, and MMP-1, correlated with age in both groups and thus may reflect normal aging processes rather than disease. Another three inflammatory markers, consisting of TGF-α, EN-RAGE, and OSM, only correlated with age in the patients and, interestingly, were also increased in the plasma from patients compared to CBD (Figure 1B and Figure 2B). The strongest correlation with age in CBD was observed for CDCP1, a molecule with a role in immune cell migration and chemotaxis [29,30,31], whereas in the patients, the strongest correlation was observed for CST5, a cysteine protease inhibitor which can also modulate gene transcription and protein expression [32,33]. The inflammatory markers that varied with age can be grouped according to function and are shown in Figure 2C for the two groups. Inflammatory markers associated with the activation of the immune cell response and apoptosis were enhanced in plasma from the patients.

### 2.4. The GABA Concentration in Plasma and Correlation of GABA or MADRS-S Score with Levels of Inflammatory Markers

Since GABA is exclusively generated within the body and is the main inhibitory neurotransmitter in the brain, we examined whether the GABA concentration in plasma varied between CBD and the patients (Figure 2D). The results showed that the GABA concentration ranged from 253 to 824 nM in the two groups and revealed a somewhat increased plasma GABA concentration in the patients, resulting in a significantly higher average plasma concentration (CBD: 586 ± 25 nM; PD: 683 ± 19 nM; *p* = 0.003). In general, the plasma GABA levels did not correlate with the levels of inflammatory markers in plasma. The exceptions were LIF-R and ST1A1 in the plasma from CBD (Figure 2E, Appendix A). Furthermore, the levels of GABA in plasma did not correlate with the age of CBD or the patients, the MADRS-S score, or the BMI of the patients. A post hoc analysis with a *t*-test showed elevated levels of GABA in the patients with benzodiazepines and/or Z-drugs when compared to the remaining patients (*t*-value -2.354, p-value 0.037). Importantly, most of these patients were also treated with other medications with potential for influencing GABA levels. None of the markers correlated with BMI. Only one marker, IL-18, correlated with the MADRS-S score of the patients (Figure 2F; Appendix A, *R*-value −0.4832, *p*-value 0.017).

### 2.5. The GABA Signaling System is Altered in PBMCs from the Patients

GABA can activate two types of receptors in the plasma membrane of cells: the GABA_A_ receptors that are chloride ion channels opened by GABA and the G-protein-coupled GABA_B_ receptor [1,34,35]. The GABA_A_ receptors are homo- or hetero-pentamers formed by a selection of subunits from 19 known isoforms (α1-6, β1-3, γ1-3, δ, ε θ, π, and ρ1-3) [35]. In contrast, the GABA_B_ receptor is normally formed as a dimer of the two isoforms identified to-date [34,36]. The ρ2 subunit was the only GABA_A_ subunit, which was expressed in PBMCs from most of the CBD and patients (Figure 3A, Table 2). The expression level was similar in both groups and could indicate the formation of homomeric ρ2 GABA_A_ receptors in the cells. Approximately 30–40% of the CBD also expressed the β1, δ, and ε subunits, while the expression of these subunits was less frequent in the patients (Table 2). Other GABA_A_ subunits were only infrequently expressed in both groups (Table 2). Only one GABA_B_ subunit was expressed in both patients and CBD, indicating that the traditional GABA_B_ receptors may not be formed in the PBMCs (Table 2).

The strength of GABA_A_ receptor signaling depends, in part, on the chloride gradient across the cell membrane. Therefore, we examined whether the expression of chloride transporters that regulate the intracellular chloride concentration differed between the CBD and the patient cells (Figure 3B). The NKCC1 transporter that transports chloride ions into the cells was significantly down-regulated in the PBMCs from the patients. The other three transporters, which move Cl^−^ out of the cells, were also expressed in the majority of samples, but at similar levels in both groups.

### 2.6. GABA Regulates the Release of Inflammatory Markers from Stimulated PBMCs from the Patients

Since the levels of inflammatory markers in plasma differed between the two groups and GABA has been reported to be immunomodulatory [7], we examined whether GABA affected the release of inflammatory markers from the patients’ PBMCs. The culture media of anti-CD3-stimulated PBMCs were assessed using the same Olink Inflammation protein panel composed of the 92 inflammatory markers that were used for an analysis of the plasma samples described above. We also examined whether GABA at concentrations of 100 and 500 nM regulated the secretion of specific inflammatory markers.

A total of 59 different inflammatory markers were detected in the culture media from both non-stimulated and anti-CD3-stimulated PBMCs (Figure 4A; Appendix A), as well as an additional nine markers (IL-2, IL-4, IL-5, IL-13, IL-2RB, IL-10RA, IL-15RA, PD-L1, and SLAM-F1) in the media from stimulated cells. The majority of inflammatory markers were secreted at a higher level in the stimulated PBMCs (Figure 4A). Only VEGF-A was secreted at a significantly higher level in the resting state of the cells (Figure 4A). From the stimulated PBMCs, 51 inflammatory markers were the same as those identified in the plasma samples, while the remaining 17, including INF-γ, TNF-α, and IL-1α, were not detected in the plasma samples. We further examined whether treatment with GABA altered the release of inflammatory markers from the stimulated PBMCs compared to non-treated PBMCs. When 100 nM GABA was applied, there was a significant increase in the secretion of two inflammatory markers, AXlN1 and TNFRSF9, and a decrease of another two, VEGF-A and IL-1α (Figure 4B, Appendix A). With 500 nM GABA treatment, the secretion of four inflammatory markers was significantly decreased from the stimulated PBMCs: CD244, IL-13, HGF, and VEGF-A (Figure 4C; Appendix A). Of the inflammatory markers regulated by GABA, HGF and VEGF-A increased with age in both CBD and the patients (Figure 1B). Interestingly, of the seven inflammatory markers regulated by GABA, only VEGF-A was modulated by both concentrations of GABA.

## 3. Discussion

### 3.1. GABA and Age Modulate the Inflammatory Environment

This study examined peripheral inflammatory markers and the immunoregulatory effects of GABA and GABA signaling in PBMCs from patients with a major depressive episode (Figure 5). The results identified thirteen inflammatory markers that were upregulated in plasma from the patients. Consistent with other studies, a number of inflammatory markers correlated with age for both CBD and the patients, but the most prominent age-associated marker differed for the two groups, being CDCP1 for the CBD and CST5 for the patients. The main GABA_A_ receptor in the PBMCs was unchanged in the patients’ PBMCs, but the NKCC1 transporter was down-regulated. Physiological concentrations of GABA modulated the release of inflammatory markers from the patients’ immune cells. The results support an immunoregulatory role of GABA-activated GABA_A_ receptor signaling in PBMCs.

### 3.2. Inflammatory Markers in Psychiatric Disorders

The inflammatory environment is thought to be altered in many psychiatric disorders [8,9,12]. The results in this report showed clear differences at the systemic level with the altered plasma concentrations of specific inflammatory markers in the patients (Figure 5). Inflammatory markers released from stimulated PBMCs from the patients identified additional markers that may be significant as auto- or paracrine signals. In the present study, the patients exhibited increased levels of neuronal growth factors (NGFs) and neurotropin-3 (NT3) compared to CBD and, notably, NGFs were not released from the simulated PBMCs. Low levels of NGFs have been implicated in the pathogenesis of depression and observed in patients with depressive disorders [37,38]. β-NGF and NT3 are the NGFs that increase the viability, growth, and development of neurons and may suppress inflammation [39]. It is possible that the increase in NGF levels in the patients is related to improvements in responses to the antidepressants. This is in accordance with studies in animal models, where increased expression/concentration levels of NT3 and NGF were observed in a number of brain regions and/or in serum in response to treatments with antidepressants [40,41,42,43]. Interestingly, a recent study of inflammatory markers altered in plasma from patients with the autoimmune disease type 1 diabetes (T1D) or secreted by stimulated T1D PBMCs, identified many of the same inflammatory markers as in this study [7]. How small molecules from the brain reach circulation is still being explored, but they may diffuse by volume transmission from the brain to the blood [27] by the route of the active glymphatic system.

A number of studies have reported alteration in inflammatory markers with medical treatments. Recent meta-analysis studies have shown decreased plasma levels of IL-6, IL-10, IL-1β, TNF-α, p11, IFN-γ, and CCL2 after treatment with antidepressants, including SSRI and SNRI [18,44,45,46]. A reduced mRNA expression of IL-6, IL-1b, and MIF in leukocytes after the treatment of patients with antidepressants, escitalopram, or nortriptyline has been reported [21]. Additionally, high levels of VEGF-A mRNA in whole blood from patients with depression were reported, although the VEGF protein levels in the plasma were not affected [47]. Further, an increased IFNγ /IL-10 ratio and changes in CCL11 and IFNγ with antidepressant treatments have been reported [48,49], but the lithium augmentation of antidepressants had no effect on the inflammatory markers in MDD [50].

Several inflammatory markers, which are often reported to be altered in patients with MDD, did not differ significantly between the patients in this study and CBD, e.g., IL-6. A part of the explanation may potentially be that the antidepressant treatment had normalized the levels of some of the inflammatory markers. However, MIP-1α (i.e., CCL3), CCL4, and CCL20, the macrophage-released pro-inflammatory molecules, were significantly elevated in the patients, consistent with previous MDD reports [51,52]. IL-18 was the only inflammatory marker that correlated with the MADRS-S score and is consistent with other studies of inflammatory markers and depression [53,54,55,56]. Many inflammatory markers correlated with age in both CBD and the patients in concordance with previous studies [57,58,59,60]. Our study corroborates that CDCP shows the strongest correlation with age in healthy individuals [59], whereas CST5 (cystatin D) has the strongest correlation with age in the patients. CST5 is an inhibitor of lysosomal and secreted cysteine proteases, but can also locate to the nucleus and modify gene transcription [33]. Interestingly, CST5 is an ultra-early biomarker of traumatic brain injury [32].

### 3.3. Plasma GABA Concentrations

The average plasma levels of GABA were increased in the patients compared to age- and sex-matched CBD GABA levels, but there was, nevertheless, a narrow range of values and considerable overlap of sample GABA concentrations observed between the two groups. Circulating levels of GABA are much lower than those found in the synaptic cleft and instead, comparable to levels activating high-affinity extrasynaptic GABA_A_ receptors that generate small-amplitude, but long-lasting, currents in neurons [1]. In previous studies where plasma GABA levels have been measured in samples from patients with psychiatric disorders and a control group, the results have varied [26,61,62,63,64,65,66]. Since the majority of patients in this study were being treated (with medicines, e.g., benzodiazepines, Li^+^, valproate, or antipsychotic medication), it is possible that the difference observed in the GABA plasma concentration is related to the effects of the medications [65,67].

### 3.4. Immunomodulation by GABA

We and others have shown that immune cells can be regulated by GABA. GABA can, e.g., decrease the proliferation of T cells, reduce inflammation in experimental autoimmune encephalomyelitis, decrease cytokine secretion from T cells, and modulate the mobility of infected dendritic cells [3,4,5,7,68,69]. The PBMCs express genes for the diverse components of the GABA signaling system, including the GABA_A_ receptors and the chloride transporters [7] and respond to GABA by activating GABA_A_ receptor channels [4,7]. NKCC1, the transporter that maintains a high intracellular chloride concentration in the cells, was down-regulated in PMBCs from the patients, implying a decreased strength of GABA signaling in the cells. The alteration in the chloride gradient across the plasma membrane is potentially partially compensated for by the somewhat increased GABA concentration in the plasma. A similar down-regulation of NKCC1 has been observed in cells from both healthy and depressed pregnant women [28] and from T1D patients [7]. This observation indicates a general shift of immune regulation by altered GABA signaling, rather than a decrease specifically associated with depression. Another possibility is that the reduced expression of NKCC1 is a trait conferring vulnerability for depression.

### 3.5. Limitations of the Present Study

This study has several limitations. First, the patients were recruited in a naturalistic setting and had different combinations of medications that may influence GABA signaling. The careful selection of patients has, however, reduced the effects of other confounders, such as alcohol use and recent electroconvulsive therapy. Secondly, patients with depression are in different stages of the disease process. Thirdly, as this is a pilot study, we have not conducted correction for multiple testing and the findings must be validated in new cohorts. Finally, here, the control blood was obtained from the hospital blood-central facility and donors were only matched for sex and age and were not evaluated in terms of mental health. A strength of this study is the inclusion of patients with severe depressive states, which increases the likelihood that these results are relevant for a clinical psychiatric population.

### 3.6. Conclusions and Further Studies

The study shows that significant changes take place in the immune system during depression and identifies molecules and mechanisms important for immunomodulation in depression (Figure 5). The levels of several inflammatory molecules, including NGFs, IL-18, and CST5, were altered in the plasma of patients with depression. In PBMCs from the patients, GABA modulated the release of cytokines. The average GABA level in the plasma from patients was increased, whereas NKCC1 expression in the PBMCs was decreased. Together, the results suggest altered GABA signaling during depression. Future studies are required to understand if specific subpopulations of immune cells are involved in mental illness and, then, how they are regulated. Further, the link between concentrations of small molecules like GABA, NGFs, and CST5 in plasma and brain functions needs to be explored.

## 4. Materials and Methods

### 4.1. Study Individuals, Ethical Permits, and Blood Samples

Psychiatric illness was diagnosed using the International Neuropsychiatric Interview (M.I.N.I. 6.0) and the Diagnostic and Statistical Manual of Mental Disorders (DSM)-IV criteria. The interviews were conducted by two resident physicians in psychiatry and a specialized nurse in psychiatry at the clinic [70]. Patients were recruited at the Department of General Psychiatry at Uppsala University Hospital, Sweden, and the inclusion criteria for this study were that they met the DSM-IV criteria for a current moderate to severe depressive episode and had either a major depressive disorder or bipolar disorder at the time of blood sample collection. Depression severity was assessed using the self-rating version of the Montgomery Åsberg depression rating scale (MADRS) [71,72]. It is a 10-item clinician-rated scale measuring the severity of depressive symptoms, including the following items: reported sadness, inner tension, reduced sleep, reduced appetite, concentration difficulties, lassitude, inability to feel, pessimism, and suicidal thoughts. The items are rated on a Likert scale from 0 to 6 and the total score ranges from 0 to 60. Higher scores indicate a greater severity. The study was approved by the Regional Ethics Committee in Uppsala (D.nr 2014/148 2014-06-12 and 2015-11-02), and all participants provided written consent. Twenty-five psychiatric patients with a major depressive episode participated in the study. Venous blood samples from the patients (PD) were collected in EDTA tubes and used to isolate plasma and PBMCs. Twenty-six control blood samples from blood donors (CBD) were obtained at the blood center at Uppsala University Hospital. CBD were age- and gender-matched to the patients, but were not evaluated in terms of mental health.

### 4.2. Plasma and PBMC Isolation from Blood Samples

Plasma and PBMCs were isolated from freshly drawn blood samples, as previously described [28]. The plasma was isolated by centrifugation at 3600 rpm for 10 min at 4 °C and immediately frozen at −80 °C. PBMCs were prepared by first diluting the blood samples in an equal volume of MACS buffer (Miltenyi Biotec, Madrid, Spain), and layered on Ficoll-paque plus (Sigma-Aldrich, Hamburg, Germany). Briefly, the samples were then subjected to density gradient centrifugation at 400 *g* for 30 min at room temperature. The PBMCs were carefully withdrawn and washed twice in MACS buffer. A portion of purified PBMCs was saved in RNAlater (Sigma-Aldrich, Hamburg, Germany) at −80 °C for mRNA extraction for qPCR experiments, and the remaining portion was used for an analysis of cell culture supernatants by a multiplex proximity extension assay (PEA).

### 4.3. Multiplex PEA for Inflammatory Marker Measurements

Plasma samples and culture media supernatants were analyzed using multiplex PEA with a panel of 92 inflammation-related proteins (multiplex Inflammation I^96 × 96^, Olink Proteomics^TM^, Uppsala, Sweden), as previously described [73]. In brief, 1 µL of sample or negative control was incubated overnight at 4 °C with a panel of oligonucleotide-conjugated antibodies, where each target protein could be recognized by a pair of antibodies. This binding brought the attached oligonucleotides in close proximity, allowing them to hybridize to each other and subsequently extend via enzymatic DNA polymerization, creating DNA amplicons, which were quantified using a microfluidic-based quantitative real-time PCR system (Fluidigm, San Francisco, CA, USA). To even out intra-plate variations, the raw quantification cycle (Cq) values were normalized against spiked-in controls and negative controls to achieve normalized protein expression (NPX) values. NPX is an arbitrary value on a log_2_ scale, where an increase of a unit corresponds to a two-fold increase of the protein concentration. These NPX data were then converted to linear data, using the formula 2^NPX^, prior to further statistical analysis. Each protein has its own limit of detection (LOD), defined as the NPX of the background plus three times the standard deviation. The multiplex PEA is reported to have a sensitivity in a subpicomolar and broad dynamic range. The technical performances, including the LOD, dynamic range, etc., for all the proteins included in the panel are available at the manufacturer’s homepage: https://www.olink.com/data-you-can-trust/publications. Proteins with levels below the LOD were excluded from further data analysis.

### 4.4. Determination of the GABA Concentration

Plasma samples were thawed and the levels of GABA were measured using an ELISA kit (LDN Labor Diagnostika Nord, Nordhorn, Germany), as per the manufacturer’s guidelines [69]. In brief, plasma samples and standards provided in the kit were extracted on an extraction plate, derivatized using equalizing reagent, and subjected to standard competitive ELISA in GABA-coated microtiter strips. The absorbance of the solution in the wells was read at 450 nm within 10 min using a Multiskan MS plate reader (Labsystems, Vantaa, Finland). The optical density was used to calculate the GABA concentration using a standard curve.

### 4.5. Real-Time Quantitative Reverse Transcription PCR

The PBMC samples collected from 26 CBD and 25 patients were subjected to total mRNA extraction using an RNA/DNA/Protein Purification Plus Kit (Norgen Biotek, Thorold, Ontario, Canada). The RNA concentration was measured using Nanodrop (Nanodrop Technologies, Thermo Scientific, Inc., Wilmington, DE, USA). Further, 1.5 μg RNA was treated with 0.6 U DNAase I (Roche, Basel, Switzerland) for 30 min at 37 °C, with 8 mM EDTA for 10 min at 75 °C, and was then converted to cDNA using Superscript IV reverse transcriptase (Invitrogen, Stockholm, Sweden) in a 20 μL reaction. A reverse transcriptase negative reaction was also carried out in order to confirm the absence of genomic DNA contamination. The gene-specific primer pairs are listed in Appendix A. The PCR amplification was performed using the ABI PRISM 7900 HT Sequence Detection System (Applied Biosystems, Gothenburg, Sweden) with an initial denaturation step of 5 min at 95 °C, followed by 45 cycles of 95 °C for 15 s, 60 °C for 30 s, and 72 °C for 1 min.

### 4.6. PBMC Supernatants for Multiplex PEA

The cells were suspended in complete medium (RPMI 1640 supplemented with 2 mM glutamine, 25 mM HEPES, 10% heat-inactivated fetal bovine serum, 100 U/mL penicillin, 10 mg/mL streptomycin, 5 µM β-mercaptoethanol, (Sigma-Aldrich, Hamburg, Germany)) in a concentration of 1 million cells per mL. The cells (100 µL = 100,000 cells) were added in triplicate for each experimental group to the 96-well plates pre-coated with 3 μg/mL anti-CD3 antibody (clone HIT3a, BD Biosciences, Allschwil, Switzerland) for 3–5 h at 37 °C. The cells were then incubated in the presence or absence of GABA at the relevant concentration for 72 h at 37 °C (95% O_2_, 5% CO_2_) and supernatant culture media were collected, centrifuged to remove cellular debris, and stored at −80 °C for the analysis of inflammatory markers using the multiplex PEA, as mentioned above.

### 4.7. Experimental Design and Statistical Rationale

Plasma samples from 25 patients and 26 age- and sex-matched controls were used for proteome analysis by multiplex PEA and for determination of the GABA concentration in plasma. Further, PBMCs isolated from the aforementioned 26 controls and 25 patients were used for mRNA expression analysis by qPCR performed in technical duplicates. Supernatants from resting and activated (non-treated vs. GABA 100 nM and GABA 500 nM treated) PBMCs from 16 patients were used for proteome analysis by multiplex PEA.

Statistical analysis and data mining were performed using Statistica 12 (StatSoft Scandinavia, Uppsala, Sweden) and GraphPad Prism 7 (La Jolla, CA, USA). The statistical tests were performed after omitting outliers identified by the Tukey test. The differences between groups were assessed by nonparametric Kruskal–Wallis ANOVA on ranks with Dunn’s post hoc test. The contingency of sex equality between the two groups was assessed by Fisher’s exact test and age was assessed by a non-parametric Mann–Whitney test. The correlation between inflammatory markers and demographic factors was assessed using non-parametric Spearman rank correlation. To reduce the risk of false discoveries caused by multiple testing, the Benjamini–Hochberg false discovery rate method was used [74]. The significance level was set to *p* < 0.05.

## Figures and Tables

**Figure 1 ijms-20-06172-f001:**
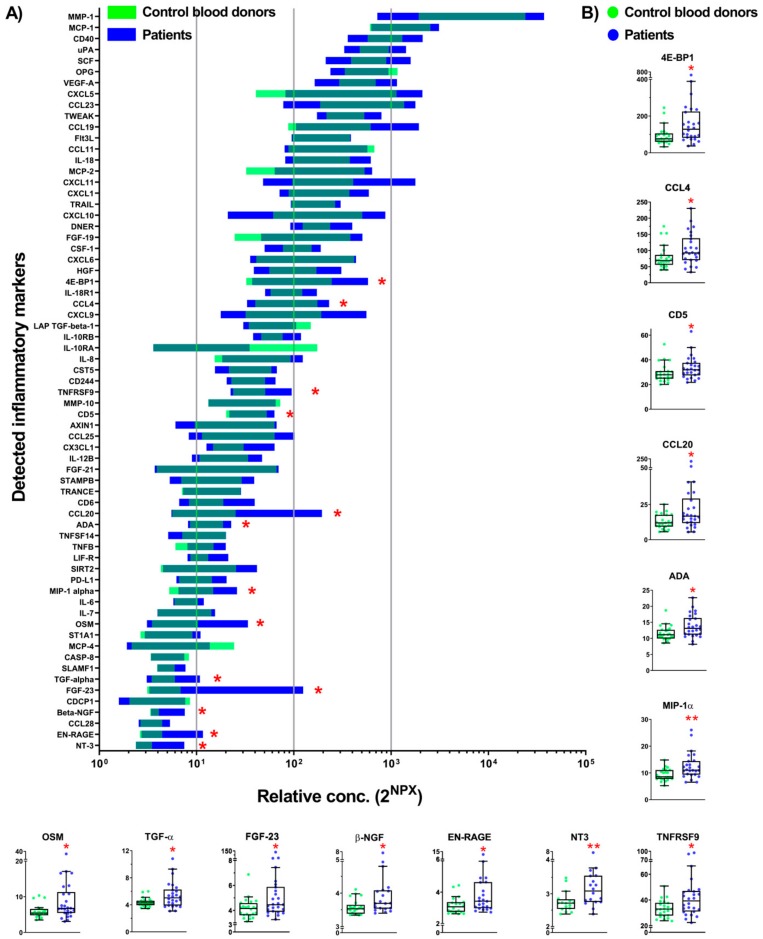
Inflammatory markers in plasma from control blood donors (CBD) and patients. (**A**) Screening of 92 inflammatory markers (Appendix A) in plasma samples from CBD (*n* = 26) and patients (*n* = 25) by Proseek Multiplex PEA inflammation panel I detected the expression of 67 markers (Appendix A). Data are presented by 2^NPX^ (Normalized Protein Expression) values as floating bars (minimum to maximum) arranged in descending order of the mean expression level of inflammatory markers. (**B**) Inflammatory markers with a significantly changed expression level in the plasma of patients compared to CBD. The differences between groups were assessed by nonparametric Kruskal–Wallis ANOVA on ranks with Dunn’s post hoc test. Data are shown as a box and whiskers overlapped with a scatter dot plot. * p < 0.05, ** p < 0.01.

**Figure 2 ijms-20-06172-f002:**
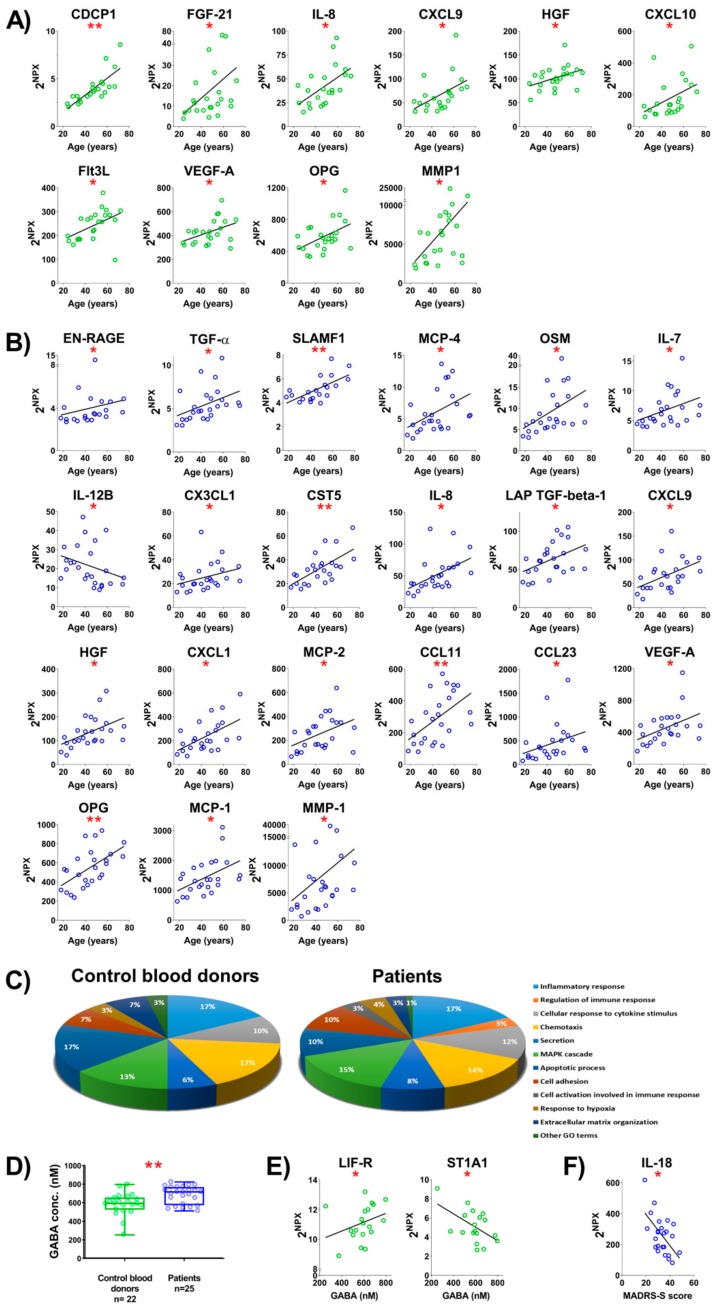
Age, gamma-aminobutyric acid (GABA), and Montgomery Åsberg depression rating scale (MADRS)-S score correlate with levels of inflammatory markers. Correlation between levels of inflammatory markers in plasma and age; (**A**) CBD and (**B**) patients. Only inflammatory markers with a statistically significant correlation are shown. (**C**) Classification based on the cellular functions of markers that were significantly correlated with age of CBD (10 inflammatory markers) and patients (21 inflammatory markers). (**D**) Quantification of GABA levels in plasma from CBD and patients. (**E**) Correlation between levels of inflammatory markers and GABA levels in plasma from CBD. (**F**) Correlation between the level of IL-18 in plasma from patients and the MADRS-S score for the patients. The correlation between inflammatory markers and demographic factors was accessed using non-parametric Spearman rank correlation. To reduce the risk of false discoveries caused by multiple testing, the Benjamini–Hochberg false discovery rate method was used. Rho values and p values of correlation statistics are provided in Appendix A. * *p* < 0.05, ** *p* < 0.01.

**Figure 3 ijms-20-06172-f003:**
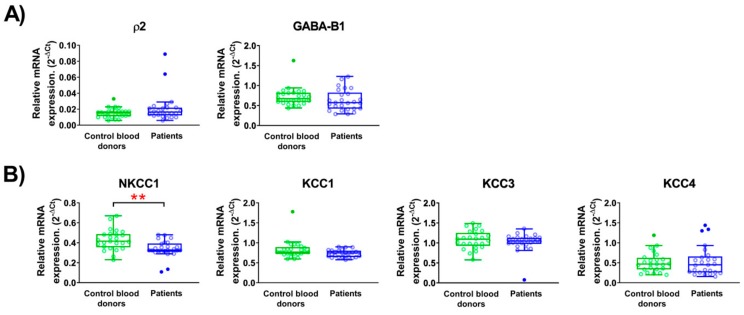
The relative mRNA expression in peripheral blood mononuclear cells (PBMCs) from CBD and patients. (**A**) GABA_A_ receptor subunit ρ2 and GABA_B_ receptor subunit B1 expression level. (**B**) Chloride co-transporters: NKCC1, KCC1, KCC3, and KCC4 expression level. Data are shown as a box and whiskers overlapped with a scatter dot plot. The outliers were detected using Tukey’s test (with 1.5 times +/− IQR, inter quartile range) and are shown with filled circles. Normality of data was assessed by the Shapiro–Wilk normality test (Appendix A). ** *p* < 0.01.

**Figure 4 ijms-20-06172-f004:**
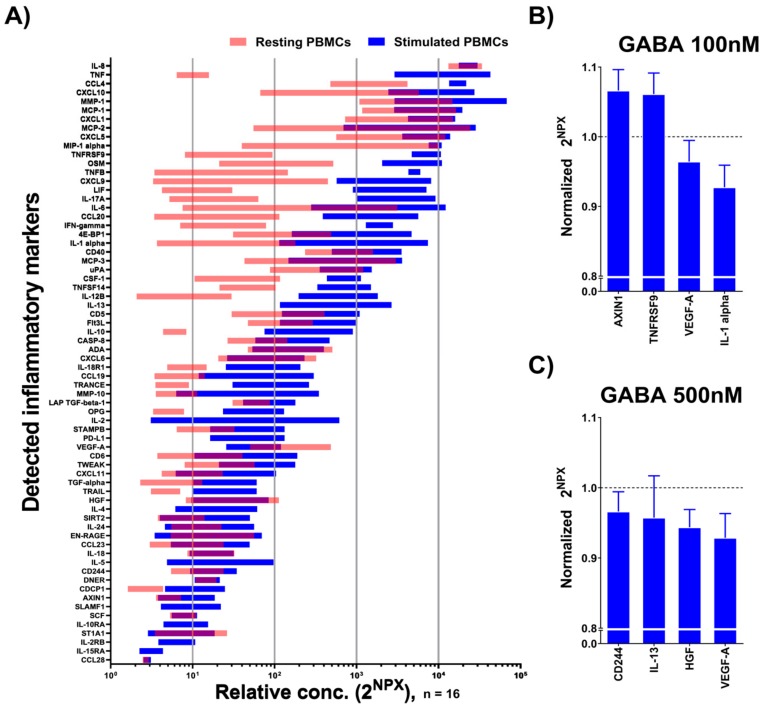
Identification of cytokines released from PBMCs from patients and the effects of GABA on the levels of inflammatory markers released from stimulated cells. (**A**) Screening of 92 inflammatory markers (Appendix A) in PBMC media from patients by Proseek Multiplex PEA inflammation panel I revealed the expression of 59 (light red) and 68 (blue) inflammatory markers from non-stimulated and stimulated PBMCs, respectively (Appendix A). Data are represented by 2^NPX^ values as floating bars (minimum to maximum) arranged in descending order of the mean expression level of the inflammatory markers. (**B**–**C**) Inflammatory markers released from stimulated PBMCs from patients were significantly affected by (**B**) GABA 100 nM or (**C**) GABA 500 nM. Data are represented by 2^NPX^ values normalized to controls as a bar graph with the mean ± SEM. Mean values with SEM and p values are provided in Appendix A. The differences between groups were assessed by nonparametric Kruskal–Wallis ANOVA on ranks with Dunn’s post hoc test (Appendix A). *p* < 0.05 for (**B**) and (**C**).

**Figure 5 ijms-20-06172-f005:**
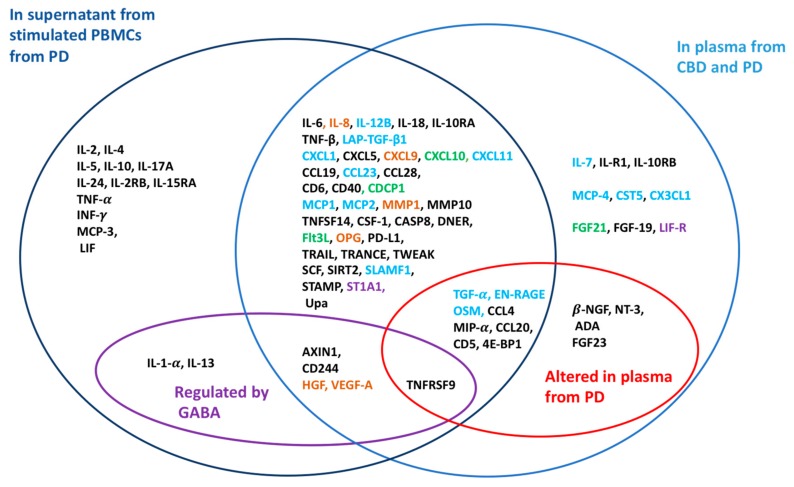
Inflammatory markers in plasma or released from stimulated PBMCs in vitro. Dark blue circle: Inflammatory markers detected in the supernatant from stimulated PBMCs from patients (PD). Light blue circle: Inflammatory markers detected in plasma samples from CBD and PD. Violet circle: Inflammatory markers regulated by GABA in PBMCs from PD. Red circle: Inflammatory markers altered in the plasma of PD compared to CBD. Blue: Inflammatory markers that correlated with age, but only in PD; green: Inflammatory markers that correlated with age, but only in CBD; brown: Inflammatory markers that correlated with age in both PD and CBD; violet: Inflammatory markers that correlated with the GABA concentration in plasma from CBD.

**Table 1 ijms-20-06172-t001:** Characteristics of patients.

**Participants** (*N*)	25
Level of care at inclusion, n (%):	
Inpatient	20 (80%)
Day program for depression	5 (20%)
Age (Mean (SD))	43.96 (15.7)
Gender (M:F)	12:13
BMI (Mean (SD))	25.3 (6.4)
**Diagnosis**: *n* (%):	
Current depressive episode	25 (100)
Major depressive disorder	20 (80)
First depressive episode	3 (12%)
Recurring unipolar depression	17 (68%)
Bipolar disorder	5 (20%)
Type I	4 (16%)
Type II or uncategorized	1 (4%)
Any anxiety disorder	7 (28%)
Other psychiatric diagnoses *	4 (16%)
Previous hospitalization for depression (*n* (%))	22 (88%)
MADRS-S score (mean (SD))	33.8 (7,4)
**Medication**, *n* (%):	
Other anxiolytic medications **	11 (44%)
Antidepressive treatment ***	21 (84%)
Antipsychotics	6 (24%)
Benzodiazepines	5 (20%)
Z-analogues	6 (24%)

* One case of Asperger’s and dyslexia, one case of ADHD, one case presented psychotic symptoms, and one patient has since this study committed suicide. ** Sedating antihistamines, phenothiazines. *** SSRI, SNRI, mood stabilizers and atypical antidepressants.

**Table 2 ijms-20-06172-t002:** The percentage of samples expressing the particular mRNA.

	CBD	Patients
**GABA_A_ Receptor Subunits**		
GABRA1 (α1)	0	0
GABRA2 (α2)	0	0
GABRA3 (α3)	3.8	4
GABRA4 (α4)	15.4	8
GABRA5 (α5)	19.2	4
GABRA6 (α6)	15.4	16
GABRB1 (β1)	30.8	8
GABRB2 (β2)	38.5	36
GABRB3 (β3)	0	0
GABRG1 (γ1)	0	4
GABRG2 (γ2)	0	0
GABRG3 (γ3)	0	0
GABRD (δ)	34.6	12
GABRE (ε)	42.3	20
GABRQ (θ)	0	0
GABRP (π)	3.8	4
GABRR1 (ρ1)	0	0
GABRR2 (ρ2)	100	96
GABRR3 (ρ3)	0	12
**GABA_B_ Receptor Subunits**		
GABBR1 (GABA-B1)	100	100
GABBR2 (GABA-B2)	0	0
**Chloride Transporters**		
SLC12A2 (NKCC1)	100	100
SLC12A1 (NKCC2)	0	0
SLC12A4 (KCC1)	100	100
SLC12A5 (KCC2)	0	0
SLC12A6 (KCC3)	100	100
SLC12A7 (KCC4)	96	100

Total of 51 PBMC samples were examined, including 26 from CBD and 25 from patients.

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
