# Peer review of "Depression, GABA, and Age Correlate with Plasma Levels of Inflammatory Markers"

_ijms, 2019, doi:10.3390/ijms20246172_

Round 1

Reviewer 1 Report

The manuscript describes correlation between depression, GABA and inflammatory markers. The study has found that inflammatory markers are elevated in depression. The finding is significant and the manuscript could be accepted for publication after minor corrections of English grammar and sentence redactions.

Author Response

We have gone through the manuscript and adjusted the English and sentence structure where appropriate.  Please see revisions highlighted by yellow in the revised manuscript.

Reviewer 2 Report

This manuscript reported correlation of depression, GABA and age with plasma levels of inflammatory markers. I have some minor comments:

1. Authors did not assess the stress response and coping skills of the patients. Stress has been shown to play a significant role in alterations of circulating inflammatory markers.

2. The controls recruited in the study are stated as age and gender matched healthy controls. Authors did not provide any details about the scales used for psychiatric assessment of the controls. 

3. Line 178-183, The strength of GABAA receptor signaling depends on the chloride gradient across the cell membrane. The activity of chloride transporter is also regulated by phosphorylation. Did author check the phosphorylation level of KCCs and NKCCs?.

Author Response

It is correct we did not investigate stress response or coping skills of the patients.We agree this is also important and do address this in the introduction of the article see page 1, line 36-37 and page 2, line 43-44.  However, effects of stress on depressed patients is important to study in a controlled manner. We agree, it should be done but in another study. This is now addressed in line 88-89, and line 379-379. Where we state “CBD were age and gender matched to the patients but were not evaluated in terms of mental health”. We. collected blood from the blood central at the Uppsala University Hospital.  We acknowledge this limitation in section 3.5 Limitations of the study of the Discussion, line 342-343. We have not examined the phosphorylation level of the KCCs and NKCC but we agree with the reviewer that it would be interesting to examine to see if it is also altered, in addition to the decreased level of the NKCC mRNA. This could be done in future studies.

We have gone through the manuscript and adjusted the English and sentence structure where appropriate.  Please see revisions highlighted by yellow in the revised manuscript.

Reviewer 3 Report

This is an interesting paper where the authors analysed the immunological markers, the expression of the GABA signalling system components in plasma from CBD and patients with a major depressive episode. The study reveals a correlation between some inflammatory markers and age, augmented levels of immunological signalling, GABA in the patient plasma and altered GABA signalling in the PBMCs from patients.   The paper is well-written and provides a significant contribution to the field

Minor comment:

In this study the authors demonstrated that thirteen inflammatory markers among 97 analyzed are significant higher in patient plasma compared to CBD. What is the relevance / importance of these cytokines in the depression? The authors should comment on and discuss these issues.

ECT and CPR should be spelled out 

Author Response

CPR and ECT are now spelled out see line 44 and line 81. In 3.2. Inflammatory Markers in Psychiatric section of the Discussion we address what the potential relevance/significance of the cytokines elevated in the patients' in plasma may be. We have now highlighted the relevant text in blue in the revised manuscript, see lines 268-276, lines 295-297, lines 300-303.

Round 2

Reviewer 2 Report

Author answered some of the my questions but others were not possible due to limitations of study. Therefore, the modified manuscript of present form is acceptable.